# GM-DDPM: Denoising diffusion probabilistic models with Gaussian Mixture Noise

## Abstract

Denoising diffusion probabilistic models (DDPM) have shown impressive performance in various domains as a class of deep generative models. In this paper, we introduce the Gaussian Mixture noise-based DDPM (GM-DDPM), which considers the Markov diffusion posterior as a Gaussian mixture model. Specifically, GM-DDPM randomly selects a Gaussian component and then add the chosen Gaussian noise, which can be demonstrated as more efficient way to perturb the signals into a simple known distribution. We further define the reverse probabilistic model as a parameterized Gaussian mixture kernel. Due to the intractability in calculating the KL divergence between Gaussian mixture models, we derive a variational bound to maximize the likelihood, offering a concise formulation for optimizing the denoising model and valuable insights for designing the sampling strategies. Our theoretical derivation highlights that *GM-DDPM only requires the inclusion of a random offset in both the diffusion and reverse processes*, which can be efficiently implemented with just several lines of code. Furthermore, we present three streamlined sampling strategies that interface with diverse fast dedicated solvers for diffusion ordinary differential equations, boosting the efficacy of image representation in the sampling phase and alleviating the issue of slow generation speed, thereby enhancing both efficiency and accuracy. Extensive experiments on benchmark datasets demonstrate the effectiveness of GM-DDPM and its superiority over the original DDPM.

## 1 Introduction

As famously pronounced by Albert Einstein, "Imagination is more important than knowledge." This maxim resonates not only in artistic and scientific realms but also within the realm of generative models. Denoising diffusion probabilistic models (DDPMs) Sohl-Dickstein et al. (2015); Ho et al. (2020); Yang et al. (2022) embody the potency of imagination in the domain of deep generative models. They have achieved tremendous success in various domains, such as image generation Dhariwal & Nichol (2021); Nichol et al. (2021); Ramesh et al. (2022); Saharia et al. (2022); Rombach et al. (2022), image restoration Kawar et al. (2022); Wang et al. (2022); Fei et al. (2023), audio and video synthesis Kong et al. (2020); Ho et al. (2022b), etc. These achievements have demonstrated the versatility and potential of diffusion models. Unlike Generative Adversarial Networks (GAN) Goodfellow et al. (2014), which rely on adversarial training, diffusion models harness the power of diffusion processes to model the data distribution. This approach circumvents issues such as mode collapse and training instability that are commonly associated with adversarial training. Furthermore, diffusion models have been shown to generate high-quality samples in a stable and effective manner. These wonderful properties make diffusion models garner extensive attention from both academia and industry.

Although diffusion models demonstrate stability in training and ability in generating high-quality images, they possess certain limitations that hinder their performance. One such limitation is their slow generation speed, primarily due to the need to traverse the whole reverse diffusion process, which involves passing through the same U-Net-based generator network hundreds or even thousands of times. To surmount this, there has been a growing interest in improving the generation speed of diffusion models, leading to the development of various fast dedicated solvers for diffusion ordinary differential equations (ODEs), such as DDIM Song et al. (2020), DPM-Solver Lu et al. (2022). These solvers achieve improved efficiency, facilitating rapid, high-quality sample

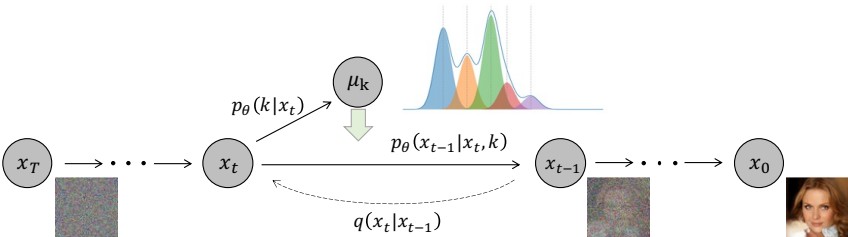

Figure 1: The illustration of the proposed GM-DDPM. In the forward process, a random offset $\mu_k$ is chosen from Gaussian means with some distribution weights $p_\theta(k|x_t)$ for $k \in [K]$.

generation. Another limitation of original diffusion models lies in their use of a single Gaussian distribution, which restricts their expressive ability to capture complex image distributions. The simplicity of a single Gaussian distribution may not be sufficient to represent the intricate structures and variations present in real-world image data. To address this limitation, we draw inspiration from natural diffusion processes and reexamine the modeling process of the original DDPM.

In natural diffusion processes, particle movement is subject to a random local force that typically conforms to a single Gaussian distribution. Accordingly, the original DDPM incorporates a single Gaussian noise in each timestep to enhance its expressive ability. In this paper, we propose a novel paradigm of DDPM called the Gaussian Mixture noise-based DDPM (GM-DDPM), which considers the Markov diffusion posterior as a Gaussian mixture model. Unlike the original DDPM Ho et al. (2020), GM-DDPM replaces the single Gaussian noise with Gaussian mixture noise, introducing multiple Gaussian components. Specifically, GM-DDPM randomly selects a Gaussian component and then add the chosen Gaussian noise, which can be demonstrated as more efficient way to perturb the signals into a simple known distribution. We further define the reverse probabilistic model as a parameterized Gaussian mixture kernel. Due to the intractability in calculating the KL divergence between Gaussian mixture models, we derive a variational bound to maximize the likelihood, offering a concise formulation for optimizing the denoising model and valuable insight for designing the sampling strategies. Our theoretical derivation highlights that GM-DDPM only requires the inclusion of a random offset in both the diffusion and reverse processes, which can be efficiently implemented with just several lines of code. The random offset is similar to adding a "random global force" in real-world diffusion, which can accelerate image degradation in the forward process, while endowing the reverse process with enhanced image reconstruction ability.

Moreover, when considering the entire diffusion process from a macroscopic perspective and applying the central limit theorem, it is suggested that this "random global force" would still approximate an additional single Gaussian distribution after a sufficient number of time steps. Therefore, we present three streamlined sampling strategies that interface with diverse fast dedicated solvers for diffusion ordinary differential equations, such as those proposed in Zhang & Chen (2022); Lu et al. (2022); Dockhorn et al. (2022). This adaptability not only allows our model to apply Gaussian mixture noise to enhance image representation in the training phase, but also solves the problem of slow generation speed, thus improving both the efficiency and precision of diffusion models. In this paper, we validate the effectiveness of GM-DDPM on benchmark datasets, demonstrating its superiority over original diffusion models under the same experimental settings. Our work contributes to pushing the boundaries of generative models and provides a promising direction for further research in diffusion modeling and its applications. Our contributions are summarized as follows:

- The paper proposes a novel framework called GM-DDPM that extends the original DDPMs by using Gaussian mixture noise to capture complex image distributions and enable more expressive image representations. We derive a variational bound to maximize the likelihood, offering a concise formulation for optimizing the denoising model and valuable insight for designing the sampling strategies.

- The paper presents three different sampling strategies that interface with diverse fast dedicated solvers for diffusion ODEs, boosting the efficacy of image representation in the training phase and alleviating the issue of slow generation speed, thereby enhancing both efficiency and accuracy.

- The paper validates the effectiveness of GM-DDPM on benchmark datasets and demonstrates its superiority over the original DDPM.

## 2    RELATED WORK

The original DDPM was first introduced by Sohl-Dickstein et al. (2015) and subsequently simplified by Ho et al. (2020). In contrast to Generative Adversarial Networks (GANs) (Goodfellow et al., 2014; Nguyen et al., 2017; Creswell et al., 2018; Lee et al., 2021; Liang et al., 2022; Wang et al., 2023), which rely on adversarial training, the original DDPM employs a diffusion process for data distribution modeling. This method involves the incorporation of two Markov chains: a forward diffusion chain and a reverse denoising chain. By introducing random perturbations at each time step, the noise is gradually diminished, culminating in the creation of high-quality samples. Since its inception, the diffusion model has been applied to various downstream tasks, leading to significant advancements in the field. Ho et al. (2022a) and Vahdat et al. (2021) proposed hierarchical architectures to stabilize the training process of diffusion models and address memory cost issues. Ramesh et al. (2022) introduced the diffusion model to text-to-image generation, achieving remarkable success with the DALL-E2 model. Rombach et al. (2022) proposed latent diffusion models that turn diffusion models into powerful and flexible generators by introducing cross-attention layers. Saharia et al. (2022) demonstrated that increasing the parameter size of the language model has a more significant impact on sample fidelity and image-text alignment than increasing the size of the image model.

Furthermore, several methods have emerged to accelerate sampling, focusing on using faster numerical ordinary differential equation (ODE) solvers (Song et al., 2020; Zhang & Chen, 2022; Lu et al., 2022; Dockhorn et al., 2022). While previous research has addressed various aspects of diffusion models, such as their application to different tasks and speed improvement techniques, the standard Gaussian distribution has poor representation ability, necessitating the use of a more uniform distribution in the latent variable space. Our work builds upon these advancements, proposing GM-DDPM to enhance the representation power of diffusion models and seamlessly adapt to fast dedicated solvers for ODEs, while also contributing novel insights and methodologies to the field.

## 3    METHOD

In this section, we introduce the proposed Gaussian-mixture noise-based diffusion denoising probabilistic models, which adds Gaussian-mixture noise to achieve more efficient diffusion process. We then identify the detail of the forward diffusion process and the reverse denoising process. We further discuss the efficiency of noise diffusion and denoising. Finally, we simplify the sampling algorithm and utilize fast ODE solvers to accelerate the sampling phase.

**Notation.**    Compared to the original diffusion model, the proposed GM-DDPM is more general to add Gaussian mixture noise rather than Gaussian noise, *i.e.*,

$$\boldsymbol{\epsilon}_t \sim \sum_{k=1}^{K} \omega_k \mathcal{N}(\boldsymbol{z}_t; \boldsymbol{\mu}_k, \sigma_k^2 \mathbf{I}), \tag{1}$$

where $\boldsymbol{\epsilon}_t$ denotes the superposition of $K$ Gaussian densities, and $\omega_k$ represents the weight of the $k^{th}$ Gaussian distribution. These weights satisfy $0 \leq \omega_k \leq 1$ and $\sum_{k=1}^{K} \omega_k = 1$. The concept of Gaussian mixture noise can be likened to a process where we initially select a Gaussian density based on the distribution weights $\{\omega_1, ..., \omega_K\}$, and subsequently, we introduce noise according to the chosen Gaussian density.

### 3.1    THE FORWARD DIFFUSION PROCESS

In the forward diffusion process, the data $\boldsymbol{x}_0$ undergoes a step-by-step corruption with Gaussian mixture noise in Eq. 1. More specifically, we define the Gaussian-mixture diffusion process as a Markov chain that gradually adds Gaussian mixture noise to the data, resulting in $\boldsymbol{x}_t = \sqrt{\alpha_t}\boldsymbol{x}_{t-1} + \sqrt{\beta_t}\boldsymbol{\epsilon}_t$. At each timestep, this diffusion process adds noise drawn from a Gaussian distribution that

is randomly chosen in $\{\mathcal{N}(\boldsymbol{\mu}_1, \sigma_1^2 \mathbf{I}), ..., \mathcal{N}(\boldsymbol{\mu}_K, \sigma_K^2 \mathbf{I})\}$ with corresponding probabilities $\omega_1, ..., \omega_K$. The Bayesian inference for the posterior $q$ is given by:

$$q(\boldsymbol{x}_{1:T}|\boldsymbol{x}_0) = q(\boldsymbol{x}_T) \prod_{t=1}^{T} q(\boldsymbol{x}_t|x_{t-1}),$$

$$q(\boldsymbol{x}_t|\boldsymbol{x}_{t-1}) = \sum_{k=1}^{K} \omega_k \mathcal{N}(\boldsymbol{x}_t; \sqrt{\alpha_t}\boldsymbol{x}_{t-1} + \sqrt{\beta_t}\boldsymbol{\mu}_k, \beta_t\sigma_k^2 \mathbf{I}). \tag{2}$$

The Gaussian-mixture diffusion process allows for closed-form sampling of $\boldsymbol{x}_t$ at any timestep $t$. Let $\overline{\alpha}_t = \prod_{i=1}^{t} \alpha_i$, $\gamma_{t,i} = \beta_i \prod_{j=i+1}^{t} \alpha_j$ for $i \in [1, t-1]$, and $\gamma_{t,t} = \beta_t$. Considering the iterates of $\boldsymbol{x}_t$, we have:

$$\boldsymbol{x}_t = \sqrt{\overline{\alpha}_t}\boldsymbol{x}_0 + \sqrt{\sum_{j=1}^{t} \gamma_{t,j}\sigma_{i_j}^2} \, \overline{\boldsymbol{z}}_t + \sum_{j=1}^{t} \sqrt{\gamma_{t,j}}\boldsymbol{\mu}_{i_j}, \tag{3}$$

where $\overline{\boldsymbol{z}}_t \sim \mathcal{N}(\boldsymbol{0}, \mathbf{I})$, and $i_j \in [K]$ denotes that the noise added at the $j$-th timestep is drawn from the $i_j$-th Gaussian distribution $\mathcal{N}(\boldsymbol{\mu}_{i_j}, \sigma_{i_j}^2 \mathbf{I})$.

In this paper, we consider a simplified form of the Gaussian mixture noise, in which each component of the noise has equal standard deviation, *i.e.*, $\sigma_1 = \cdots \sigma_K = 1$. Then, we obtain

$$q(\boldsymbol{x}_t|\boldsymbol{x}_0) = \sum_{i_1,...,i_t \in [K]} \omega_{i_1} \ldots \omega_{i_t} \mathcal{N}(\boldsymbol{x}_t; \sqrt{\overline{\alpha}_t}\boldsymbol{x}_0 + \sum_{j=1}^{t} \sqrt{\gamma_{t,j}}\boldsymbol{\mu}_{i_j}, \sum_{j=1}^{t} \gamma_{t,j}\mathbf{I}). \tag{4}$$

This shows that the Gaussian distributions at any arbitrary timestep $t$ have the same standard deviation. Moreover, if we define $\alpha_t = 1 - \beta_t$, then $\overline{\alpha}_t = \prod_{i=1}^{t} \alpha_i = \prod_{i=1}^{t}(1 - \beta_i)$, $\gamma_{t,j} = \beta_j \prod_{k=j+1}^{t} \alpha_k = \beta_j \prod_{k=j+1}^{t}(1 - \beta_k)$. This yields $\gamma_t = \sum_{j=1}^{t} \gamma_{t,j} = \beta_t + \alpha_t \gamma_{t-1}$ and $\gamma_{t+1,j} = \alpha_{t+1}\gamma_{t,j}$.

**Remark.** In contrast to the original diffusion model, the proposed GM-DDPM introduces an additional mean term $\mu$, into the forward diffusion process. When $K = 1$ and $\mu = 0$, GM-DDPM is equivalent to the original DDPM. As such, our model represents a more general paradigm compared to the original DDPM. This extended framework incorporates multiple parameterized Gaussian mixture kernels, enhancing the expressiveness of diffusion models at each timestep.

### 3.2 THE REVERSE DENOISING PROCESS

The reverse denoising process is defined as a Markov chain with learned Gaussian mixture transitions starting at $p(\boldsymbol{x}_T)$, as shown in Figure 1:

$$p_\theta(\boldsymbol{x}_{0:T}) = p(\boldsymbol{x}_T) \prod_{t=1}^{T} p_\theta(\boldsymbol{x}_{t-1}|\boldsymbol{x}_t),$$

$$p_\theta(\boldsymbol{x}_{t-1}|\boldsymbol{x}_t) = \sum_{k=1}^{K} p_\theta(k|\boldsymbol{x}_t)p_\theta(\boldsymbol{x}_{t-1}|\boldsymbol{x}_t, k), \tag{5}$$

$$p_\theta(\boldsymbol{x}_{t-1}|\boldsymbol{x}_t, k) = \mathcal{N}(\boldsymbol{x}_{t-1}; \boldsymbol{\mu}_\theta(\boldsymbol{x}_t, t, k), \Sigma_\theta(\boldsymbol{x}_t, t, k)).$$

Training is performed by minimizing a variational lower bound (VLB) of the negative log-likelihood, which is given by:

$$L_{\text{vlb}} = -\log p_\theta(\boldsymbol{x}_0|\boldsymbol{x}_1) + \sum_{t>1} \mathbb{E}_{i_{1:t-1}|\boldsymbol{x}_t, \boldsymbol{x}_0} \Big[ D_{\text{KL}}(q(i_t|\boldsymbol{x}_t, \boldsymbol{x}_0, i_{1:t-1}) || p_\theta(i_t|\boldsymbol{x}_t)) \Big]$$

$$+ \sum_{t>1} \mathbb{E}_{i_{1:t}|\boldsymbol{x}_t, \boldsymbol{x}_0} \Big[ D_{\text{KL}}(q(\boldsymbol{x}_{t-1}|\boldsymbol{x}_t, \boldsymbol{x}_0, i_{1:t}) || p_\theta(\boldsymbol{x}_{t-1}|\boldsymbol{x}_t, i_t)) \Big] + D_{\text{KL}}(q(\boldsymbol{x}_T|x_0) || p(\boldsymbol{x}_T)).$$

$$\tag{6}$$

Note that, except for the first term, each term of Eq. (6) is a KL divergence between two distributions. Also, $D_{\mathrm{KL}}(q(\boldsymbol{x}_T|x_0)||p(\boldsymbol{x}_T))$ does not depend on $\theta$, but it will be close to zero if the forward noising process adequately destroys the data distribution. Thus, we need to focus on the second and third terms in Eq. (6), which split the loss function of $t$ timesteps into two parts so that they can be trained independently.

To train a classifier that can determine which Gaussian distribution is selected from the entire Gaussian mixture distribution given $\boldsymbol{x}_t$, we aim to minimize the second term. The goal of the third term is similar to that of the original diffusion models (see Appendix A.3 for details), which is to train a denoiser. The posterior $q(\boldsymbol{x}_{t-1}|\boldsymbol{x}_t, \boldsymbol{x}_0, i_{1:t})$ and prior $p_\theta(\boldsymbol{x}_{t-1}|\boldsymbol{x}_t, i_t)$ can be expressed as Gaussians, allowing the KL divergences to be calculated with closed-form expressions instead of high-variance Monte Carlo estimates. Specifically, we have:

$$\mathbb{E}_{i_{1:t}|\boldsymbol{x}_t,\boldsymbol{x}_0}D_{\mathrm{KL}}(q(\boldsymbol{x}_{t-1}|\boldsymbol{x}_t,\boldsymbol{x}_0,i_{1:t})||p_\theta(\boldsymbol{x}_{t-1}|\boldsymbol{x}_t,i_t))$$
$$=\mathbb{E}_{i_t|\boldsymbol{x}_t,\boldsymbol{x}_0}\left[\frac{\|\mathbb{E}_{i_{1:t-1}|\boldsymbol{x}_t,\boldsymbol{x}_0,i_t}(\boldsymbol{\mu}(\boldsymbol{x}_t,\boldsymbol{x}_0,i_t|i_{1:t-1}))}{2\sigma_t^2(i_t)}-\boldsymbol{\mu}_\theta(\boldsymbol{x}_t,t,i_t)\|_2^2+\frac{\sigma_t^2}{2\sigma_t^2(i_t)}+\log\frac{\sigma_t(i_t)}{\sigma_t}\right]+C. \tag{7}$$

To represent the mean $\boldsymbol{\mu}_\theta(\boldsymbol{x}_t,t)$, we propose a specific parameterization motivated by the following analysis. We can expand the above equation further by reparameterizing $\boldsymbol{x}_t(\boldsymbol{x}_0,\boldsymbol{\epsilon},i_1,...,i_t)=\sqrt{\overline{\alpha_t}}\boldsymbol{x}_0+\sqrt{\gamma_t}\boldsymbol{\epsilon}+\sum_{j=1}^t\sqrt{\gamma_{t,j}}\boldsymbol{\mu}_{i_j}$, i.e., $\boldsymbol{x}_0=\frac{1}{\sqrt{\overline{\alpha_t}}}\left(\boldsymbol{x}_t-\sqrt{\gamma_t}\boldsymbol{\epsilon}-\sum_{j=1}^t\sqrt{\gamma_{t,j}}\boldsymbol{\mu}_{i_j}\right)$. We can then obtain the expression of $\boldsymbol{\mu}(\boldsymbol{x}_t,\boldsymbol{x}_0,i_t|i_{1:t-1})$ as:

$$\boldsymbol{\mu}(\boldsymbol{x}_t,\boldsymbol{x}_0,i_t|i_{1:t-1})=\frac{1}{\sqrt{\alpha_t}}\left(\boldsymbol{x}_t-\sqrt{\beta_t}\boldsymbol{\mu}_{i_t}-\frac{\beta_t\sqrt{\gamma_t}}{\beta_t+\alpha_t\gamma_{t-1}}\boldsymbol{\epsilon}\right). \tag{8}$$

The above equations reveal that in order to predict Eq. (8) given $\boldsymbol{x}_t$, it is necessary for $\boldsymbol{\mu}_\theta$ to satisfy certain conditions. As $\boldsymbol{x}_t$ serves as an input to the model, we can opt for the following parameterization of $\boldsymbol{\mu}_\theta(\boldsymbol{x}_t,t,i_t)$:

$$\boldsymbol{\mu}_\theta(\boldsymbol{x}_t,t,i_t)=\frac{1}{\sqrt{\alpha_t}}\left(\boldsymbol{x}_t-\sqrt{\beta_t}\boldsymbol{\mu}_{i_t}-\frac{\beta_t\sqrt{\gamma_t}}{\beta_t+\alpha_t\gamma_{t-1}}\boldsymbol{\epsilon}_\theta(\boldsymbol{x}_t,t)\right), \tag{9}$$

where $\boldsymbol{\epsilon}_\theta$ is a function approximation intended to predict $\boldsymbol{\epsilon}$ from $\boldsymbol{x}_t$. This enables us to train the mean function approximator $\boldsymbol{\mu}_\theta$ either to predict $\boldsymbol{\mu}$ or, by adjusting its parameterization, to predict $\boldsymbol{\epsilon}$. At the same time, by using Langevin dynamics Welling & Teh (2011), we can gradually sample the image from Gaussian mixture noise. In Appendix A.4, we present the comprehensive training process (as outlined in Algorithm 1) and the sampling procedure (as described in Algorithm 2), which can be efficiently implemented with just several extra lines of code.

### 3.3 THE EFFICIENCY OF DIFFUSION AND DENOISING

After presenting the framework of GM-DDPM, we further conduct a comparative analysis with the original diffusion model, highlighting the superior efficiency of the noise diffusion and denoising processes.

Specifically, we firstly perform a detailed analysis and comparison of the subjective quality and objective PSNR metric of the images from a low-level perspective in Fig. 2 and Fig. 3-(b). Our GM-DDPM exhibits a faster degradation of the images and a more significant decrease in PSNR compared to the original DDPM. In addition, in Fig. 3-(a), we showcase an experiment conducted on the CIFAR-10 dataset

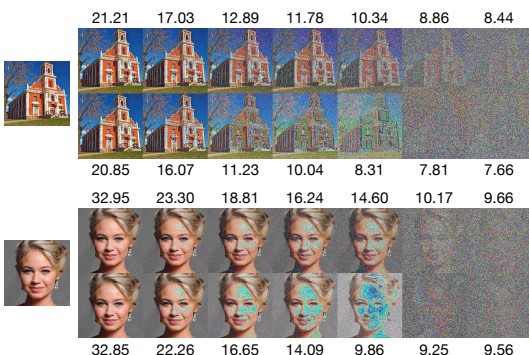

Figure 2: PSNR in the diffusion process.

Krizhevsky et al. (2009) from a high-level perspective, where noise is added to the images at different timesteps of the diffusion process. The resulting images are then fed into a pre-trained classifier, and the classification accuracy is compared. Our method demonstrates a faster decline in accuracy

compared to the original DDPM, clearly indicating the enhanced efficiency of our noise introduction strategy. The ability of our method to introduce noise more efficiently at each timestep of the forward process allows us to train a more powerful denoiser during the training phase. Consequently, during the sampling process, we can leverage the corresponding reverse process to more rapidly remove the noise, resulting in the generation of higher-quality images. This efficient noise removal capability further enhances the overall performance and fidelity of our GM-DDPM in generating realistic and high-quality images.

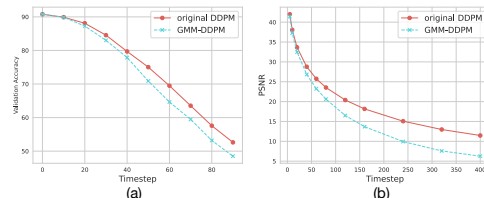

Figure 3: (a) Classification accuracy in the diffusion process on CIFAR-10 dataset; (b) PSNR in the diffusion process on AFHQ-cat dataset.

### 3.4 SIMPLIFY THE SAMPLING PROCESS

In the training phase(Algorithm 1 as shown in Appendix A.4), $\epsilon_\theta$ represents the diffusion process in our framework, while $p_\theta(i_t|\boldsymbol{x}_t)$ serves as a classifier to distinguish the mean value of noise added at each timestep. However, we encounter difficulties in directly training this classifier through experiments. As $T$ becomes larger, and noises with varying mean values are randomly added in each round, it becomes challenging to distinguish the mean and order of noise addition in rounds 1 to $t-1$ solely based on $x_t$. If the classifier is too simple, the accuracy of classification tends to be directly sampled according to the probability of noise. On the other hand, if it is too complex, it may considerably increase the network's complexity and training difficulty, surpassing that of the diffusion model training, rendering the problem meaningless. Therefore, finding an optimal balance between classifier complexity and accuracy is crucial.

To overcome these challenges, we propose three different sampling strategies to simplify the sampling process. Among them, the first strategy utilizes the original DDPM sampler Ho et al. (2020) as a foundation. In Eq. 3, in addition to the terms involving $\boldsymbol{x}_0$ and the noise $\bar{\boldsymbol{z}}t$, which are similar to the original DDPM, we introduce an additional term representing the mean value $\boldsymbol{\mu} i_j$. From a global perspective, we can know from the central limit theorem that it is equivalent to introducing a Gaussian noise globally, augmenting the original DDPM in the training phase. This augmentation enhances the model's expressiveness by incorporating additional noise sources, and theoretically, it can achieve better results by using the original DDPM sampler directly in the sampling phase. The second strategy involves directly sampling the image removing the mean value $\boldsymbol{\mu}$. We have made adjustments to the initialization sampling step based on the first strategy, enhancing its effectiveness and adaptability. The third strategy involves randomly sampling the mean value $\boldsymbol{\mu}$ according to its probability distribution and then using standard sampling to generate the image (See Appendix A.4 for detailed algorithm).

These strategies circumvent the need to estimate the mean value $\boldsymbol{\mu}$, thereby enhancing the efficiency of training and sampling, and improving the efficiency of the framework, which can be used for various tasks in computer vision and machine learning. Furthermore, since the original DDPM sampler can be used directly, our GM-DDPM seamlessly adapts to various fast dedicated solvers Song et al. (2020); Lu et al. (2022), designed for solving diffusion ODEs. This adaptability not only enhances the overall efficiency of our approach but also allows for smooth integration with existing diffusion modeling techniques.

## 4 EXPERIMENTS

This section provides a detailed overview of the implementation details of our GM-DDPM and evaluates its performance against existing methods. Firstly, we conduct a comprehensive comparison with SOTA methods. And Next, we demonstrate the plug-and-play capability of our GM-DDPM by utilizing fast dedicated solvers for diffusion ordinary differential equations (ODEs). Finally, we perform an ablation analysis to explore the impact of different Gaussian means ($\mu$) on the results of our method.

Table 1: CIFAR-10 and CIFAR-100 results.

| Methods | CIFAR-10 | | CIFAR-100 | |
|---|---|---|---|---|
| | IS ↑ | FID↓ | IS ↑ | FID↓ |
| PixelCNN Van den Oord et al. (2016) | 4.60 | 65.93 | - | - |
| SNGAN Miyato et al. (2018) | 8.22 | 21.70 | 4.66 | 90.04 |
| EBM Du & Mordatch (2019) | 6.78 | 38.20 | - | - |
| NCSN Song & Ermon (2019) | 8.87 | 25.32 | 5.63 | 110.99 |
| NCSNv2 Song & Ermon (2020) | 8.40 | 10.87 | - | - |
| ViTGAN Lee et al. (2021) | 9.30 | 6.66 | - | - |
| Glow Kingma & Dhariwal (2018) | 3.92 | 48.9 | - | - |
| DDPM Ho et al. (2020) | 9.51 | 3.35 | 10.23 | 8.79 |
| Ours(original) | 9.51 | **2.97** | 10.55 | 7.89 |
| Ours(removal $\mu$) | 9.56 | 3.00 | 9.56 | **7.80** |
| Ours(random $\mu$) | **9.58** | **2.97** | **10.65** | 10.89 |

## 4.1 COMPARISON WITH SOTA METHODS

We compare the performance of the proposed GM-DDPM with the original DDPM on several benchmark datasets, including CIFAR-10, CIFAR-100 Krizhevsky et al. (2009), ImageNet-64 Deng et al. (2009); Van den Oord et al. (2016), CelebA Liu et al. (2015), and AFHQ-v2 Choi et al. (2020). For all datasets, we set $T = 1000$. For the Gaussian mixture noise, we set $\mu = \lambda\mu_{base}$, where $\mu_{base} = \{0, +1, -1, +\sqrt{2}, -\sqrt{2}, +\sqrt{3}, -\sqrt{3}\}$, and $\lambda$ is a parameter that corresponds to different levels of noise corruption introduced during the diffusion process. Then, we set the probability $\omega$ of different means of mixture Gaussian noise as $\{0.4, 0.15, 0.15, 0.1, 0.1, 0.05, 0.05\}$. For all experiments in this section, we used a UNet model architecture similar to that used by Ho et al. (2020). We evaluated the models based on several metrics: Inception Score (IS) Salimans et al. (2016), Fréchet Inception Distance (FID) Heusel et al. (2017), and Precision-Recall (P/R) Kynkäänniemi et al. (2019). Our experiment is implemented on PyTorch with 8 Tesla v100 GPUs.

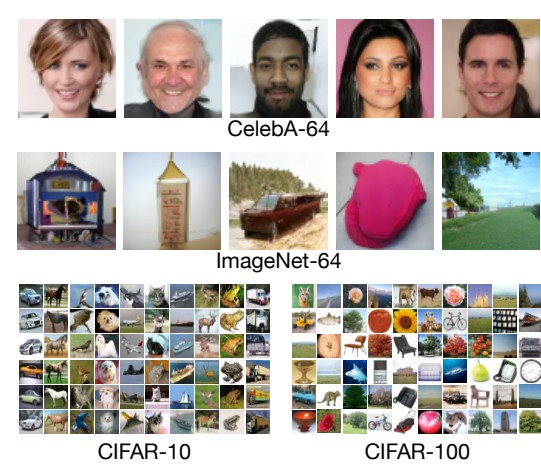

Figure 4: Several images generated on CIFAR-10 ($32 \times 32$), CIFAR-100 ($32 \times 32$), ImageNet-64 ($64 \times 64$) and CelebA-64 ($64 \times 64$) datasets.

We conduct extensive experiments to compare the performance of GM-DDPM with six existing generative models Van den Oord et al. (2016); Miyato et al. (2018); Du & Mordatch (2019); Song & Ermon (2019); Ho et al. (2020), including the original DDPM Ho et al. (2020), on the CIFAR-10 and CIFAR-100 datasets Krizhevsky et al. (2009). Table 1 presents the results of this comparison, where we evaluated the models using IS and FID metrics on both datasets. The last three rows show the results of the three sampling strategies proposed in this work. Specifically, the third-to-last row corresponds to the sampling strategy directly using the DDPM sampler (the first strategy), the second-to-last row corresponds to the sampling strategy removing the mean value $\mu$ (the second strategy), and the last row corresponds to the sampling strategy with random $\mu$ (the third strategy). GM-DDPM consistently outperforms the existing generative models across both datasets, achieving higher IS and lower FID scores. This demonstrates the superior capability of GM-DDPM in generating high-quality and diverse samples.

In addition to quantitative evaluations using IS and FID metrics, we also showcase the subjective visual quality of the generated images produced by our GM-DDPM. We present a set of sample images from several benchmark datasets, as shown in Fig. 5 and Fig. 4. To be precise, in Figure 5, we employ the sampling strategy involving random $\mu$ (the third strategy) to produce multiple $256 \times 256$ images using the AFHQ-v2 Choi et al. (2020) and CelebA Liu et al. (2015) datasets. Additionally, Fig. 4 illustrates the generated image results utilizing distinct sampling strategies. The strategy removing the mean value $\mu$ (the second strategy) is applied to the CelebA-64 Liu et al. (2015) and

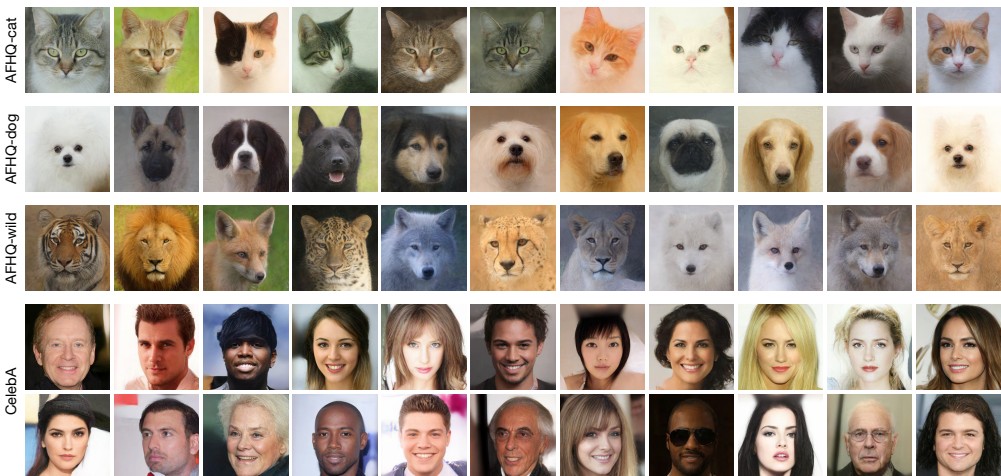

Figure 5: Several images generated on CelebA ($256 \times 256$) and AFHQ-v2 ($256 \times 256$) datasets.

ImageNet-64 Deng et al. (2009); Van den Oord et al. (2016) datasets, both of which have $64 \times 64$ resolutions. Meanwhile, the strategy that directly employs the DDPM sampler (the first strategy) is employed with the CIFAR-10 and CIFAR-100 Krizhevsky et al. (2009) datasets, which consist of $32 \times 32$ images. More results are presented in the Appendix A.5. The visual comparisons provide compelling evidence of GM-DDPM's ability to capture complex image distributions and generate visually pleasing and diverse samples, emphasizing its potential for various image generation applications and distribution modeling tasks.

Overall, the experimental results provide strong evidence of the superiority of GM-DDPM over existing generative models, highlighting its potential for various applications in image generation.

## 4.2 COMPARISON ON SOLVERS FOR DIFFUSION ODEs

We conduct comprehensive analyses to assess the superiority of our GM-DDPM over the original DDPM in experiments utilizing fast dedicated solvers for diffusion ODEs. Specifically, we compare the results between DDPM and GM-DDPM using two popular solvers, DDIM Song et al. (2020) and DPM-Solver Lu et al. (2022).

We first perform quantitative analysis using well-established evaluation metrics, including IS Salimans et al. (2016), FID Heusel et al. (2017), and Precision-Recall (P/R) Kynkäänniemi et al. (2019), to measure the quality and diversity of the generated images. The results, as shown in Table 2, consistently demonstrate that our GM-DDPM outperforms the original DDPM across multiple datasets. Our approach achieved higher IS scores and lower FID scores, indicating that GM-DDPM generates more diverse and higher-quality images. We also present more qualitative generated results in the Appendix A.5; our GM-DDPM seamlessly utilizes these solvers and consistently demonstrates higher visual fidelity, sharper details, and more diverse variations compared to the samples generated by the original DDPM. The images generated by GM-DDPM exhibit better preservation of image content and structures, indicating that our approach more effectively models complex image distributions.

These experimental findings highlight the effectiveness of our method in seamlessly utilizing solver plug-and-play, significantly accelerating the sampling process. Moreover, our GM-DDPM consistently generates more realistic and diverse images, all trained under the same experimental settings as the original DDPM. Overall, these results confirm the superiority of our GM-DDPM over the original DDPM and demonstrate its potential for a wide range of applications in image generation and distribution modeling.

## 4.3 ABLATION STUDY

In addition to the main experiments, we conduct a series of ablation studies to further investigate the performance of our GM-DDPM under different noise levels. In these ablation experiments, we

Table 2: CIFAR-10 and CIFAR-100 results using the fast dedicated solvers for diffusion ODEs.

| | Methods | CIFAR-10 | | CIFAR-100 | | AFHQ-cat | | | AFHQ-dog | | |
|---|---|---|---|---|---|---|---|---|---|---|---|
| | | IS ↑ | FID ↓ | IS ↑ | FID ↓ | FID ↓ | P ↑ | R ↑ | FID ↓ | P ↑ | R ↑ |
| DDIM | DDPM | 9.44 | 6.50 | 9.88 | 10.24 | 26.30 | **0.82** | 0.03 | 44.57 | 0.71 | 0.11 |
| | Ours | **9.52** | **5.24** | **9.98** | **9.90** | **25.06** | **0.82** | **0.04** | **44.17** | **0.73** | **0.12** |
| DPM-Solver | DDPM | 9.46 | 5.25 | 9.91 | **8.31** | 25.50 | **0.83** | **0.04** | 44.68 | 0.72 | 0.12 |
| | Ours | **9.56** | **5.10** | **10.12** | 8.42 | **23.79** | 0.82 | **0.04** | **41.20** | **0.73** | **0.13** |

Table 3: Results of ablation study.

| | Methods | CIFAR-10(lr=2e-4) | | CIFAR-10(lr=1e-4) | | CIFAR-100 | |
|---|---|---|---|---|---|---|---|
| | | IS ↑ | FID ↓ | IS ↑ | FID ↓ | IS ↑ | FID ↓ |
| $\lambda = 0$ | original DDPM | 9.29 | 3.42 | 9.51 | 3.35 | 10.23 | 8.79 |
| $\lambda = 0.1$ | Ours(original) | 9.20 | 3.60 | 9.49 | 3.31 | 10.16 | 8.78 |
| | Ours(removal $\mu$) | 9.25 | 3.68 | 9.50 | 3.29 | 10.08 | 8.93 |
| | Ours(random $\mu$) | 9.24 | 3.60 | 9.39 | 3.33 | 10.21 | 8.67 |
| $\lambda = 0.2$ | Ours(original) | 9.34 | 3.47 | 9.51 | **2.97** | 10.12 | 8.30 |
| | Ours(removal $\mu$) | **9.39** | **3.33** | 9.56 | 3.00 | 10.23 | 8.36 |
| | Ours(random $\mu$) | 9.28 | 3.42 | **9.58** | **2.97** | 10.19 | 8.29 |
| $\lambda = 1.0$ | Ours(original) | 9.31 | 4.39 | 9.57 | 4.32 | 10.55 | 7.89 |
| | Ours(removal $\mu$) | 9.33 | 4.37 | 9.55 | 4.55 | **10.84** | **7.80** |
| | Ours(random $\mu$) | 9.31 | 4.27 | 9.47 | 7.79 | 10.65 | 10.89 |

varify the noise levels added during the diffusion process and compared the results in terms of IS Salimans et al. (2016) and FID Heusel et al. (2017) metrics.

As shown in Table 3, we analyze the IS and FID of our GM-DDPM on the CIFAR-10 and CIFAR-100 datasets Krizhevsky et al. (2009) under different noise levels ($\lambda \in \{0, 0.1, 0.2, 1.0\}$), corresponding to varying degrees of noise corruption during the diffusion process. When $\lambda = 0$, it is a special case of GM-DDPM where the means $\mu$ of all Gaussian distributions are 0, which is equivalent to the original DDPM. For each noise level, we train GM-DDPM and conduct sampling using all of the sampling algorithms.

The results of the ablation study reveal several key insights. Firstly, for more complex datasets, higher expressive power is often required, resulting in better results at higher $\lambda$ values, such as $\lambda = 0.2$ on the CIFAR-10 dataset and $\lambda = 1.0$ on the CIFAR-100 dataset. At each timestep in the model, we are able to model the image more effectively, resulting in improved performance with higher noise levels. However, it is important to note that there is a limit to how much we can increase the noise level. Excessive noise levels can make the intermediate distribution network difficult to train, leading to suboptimal results. Therefore, the appropriate Gaussian mixture noise can improve the expressive ability of the model.

In summary, the ablation analysis on different noise levels reaffirms the effectiveness and robustness of our proposed GM-DDPM. It showcases the model's capability to adapt and perform well under various noise conditions, further validating its potential for various image generation tasks and distribution modeling.

## 5  CONCLUSION

In this paper, we presented GM-DDPM, a novel paradigm that extends original diffusion models by incorporating Gaussian mixture noise, thereby enhancing their ability to capture complex image distributions. By introducing multiple Gaussian components, GM-DDPM facilitates faster and more effective degradation of image structure during the diffusion process. This augmentation, combined with our proposed sampling strategies and the utilization of fast dedicated solvers, improves the efficiency and accuracy of distribution estimation. Through extensive experiments on benchmark datasets, we have demonstrated the effectiveness and superiority of our GM-DDPM. Our approach enables more expressive image representations and achieves outstanding performance in terms of image generation.

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

# A APPENDIX

## A.1 THE ORIGINAL DDPM

We briefly review the background on the vanilla diffusion models Sohl-Dickstein et al. (2015); Ho et al. (2020) which is proposed by Sohl-Dickstein et al. Sohl-Dickstein et al. (2015) and later simplified by Ho et al. Ho et al. (2020).

A diffusion model encompasses two essential components: a forward diffusion process and a reverse denoising process, both of which are defined as Markov chains. Firstly, given a data distribution $x_0 \sim q(x_0)$, we define the diffusion process $q$ which gradually destroys the data by adding Gaussian noise with variance $\beta_t \in (0, 1)$ and produces latent codes $x_1$ through $x_T$ over $T$ timesteps Nichol & Dhariwal (2021); Ho et al. (2022a); Yang et al. (2022); Croitoru et al. (2022). The diffusion process $q$ is shown as follows:

$$q(x_{1:T}|x_0) = q(x_T) \prod_{t=1}^{T} q(x_t|x_{t-1}),$$
$$q(x_t|x_{t-1}) = \mathcal{N}(x_t; \sqrt{\alpha_t}x_{t-1}, \beta_t \mathbf{I}). \tag{10}$$

where $\mathcal{N}(.)$ is a Gaussian distribution, and $\beta_t$ is noise variance which can be held constant as hyperparameters or learned by reparameterization Kingma & Welling (2013); Ho et al. (2020). Given a large enough $T$ and an apposite schedule of $\beta_t$, the distribution of the final latent code $x_T$ is close to a standard Gaussian distribution. Trough the recursive formulation (Eq. 10), the diffusion process allows sampling $x_t$ at an arbitrary timestep $t$ directly with the notation $\alpha_t := 1 - \beta_t$ and $\bar{\alpha}_t := \prod_{i=0}^{t} \alpha_i$,

$$q(x_t|x_0) = \mathcal{N}(x_t; \sqrt{\bar{\alpha}_t}x_0, (1 - \bar{\alpha}_t)\mathbf{I}) \tag{11}$$

And by sampling a Gaussian vector $\epsilon \sim \mathcal{N}(0, \mathbf{I})$, we can directly get a sample $x_t$ conditioned on the input $x_0$,

$$x_t = \sqrt{\bar{\alpha}_t}x_0 + \sqrt{(1 - \bar{\alpha}_t)}\epsilon \tag{12}$$

Similarly, the denoising reverse process is fixed to a Markov chain that gradually removes the noise therein. We can utilize a neural network to approximate the reverse distribution $q(x_{t-1}|x_t)$ which depends on the data distribution as follows:

$$p_\theta(x_{t-1}|x_t) = \mathcal{N}(x_{t-1}; \mu_\theta(x_t, t), \Sigma_\theta(x_t, t)) \tag{13}$$

And in this way, we can directly sample $x_T \sim \mathcal{N}(0, \mathbf{I})$ and utilize the estimated inverse distribution $p_\theta(x_{t-1}|x_t)$ to get image samples.

The diffusion model can be trained by minimizing a variational lower bound (VLB) of the negative log-likelihood, which is shown as follows,

$$L_{\text{vlb}} = \mathbb{E}_q[-\log p_\theta(\boldsymbol{x}_0|\boldsymbol{x}_1) + \sum_{t>1} D_{\text{KL}}(q(\boldsymbol{x}_{t-1}|\boldsymbol{x}_t,\boldsymbol{x}_0)||p_\theta(\boldsymbol{x}_{t-1}|\boldsymbol{x}_t)) + D_{\text{KL}}(q(\boldsymbol{x}_T|\boldsymbol{x}_0)||p(\boldsymbol{x}_T))$$

(14)

where $D_{\text{KL}}$ is the Kullback-Leibler divergence between two probability distributions. In Eq. 14, we can calculate the posterior $q(\boldsymbol{x}_{t-1}|\boldsymbol{x}_t,\boldsymbol{x}_0)$ by using Bayes theorem,

$$q(\boldsymbol{x}_{t-1}|\boldsymbol{x}_t,\boldsymbol{x}_0) = \mathcal{N}(\boldsymbol{x}_{t-1};\bar{\mu}(\boldsymbol{x}_t,\boldsymbol{x}_0),\bar{\beta}_t\mathbf{I})$$

(15)

where $\bar{\beta}_t = \frac{1-\bar{\alpha}_{t-1}}{1-\bar{\alpha}_t}\beta_t$, and $\bar{\mu}(\boldsymbol{x}_t,\boldsymbol{x}_0)$ is shown as follows,

$$\bar{\mu}(\boldsymbol{x}_t,\boldsymbol{x}_0) = \frac{\sqrt{\bar{\alpha}_{t-1}}\beta_t}{1-\bar{\alpha}_t}\boldsymbol{x}_0 + \frac{\sqrt{\alpha_t}(1-\bar{\alpha}_{t-1})}{1-\bar{\alpha}_t}x_t$$

(16)

To estimate $L_{\text{vlb}}$, we can utilize the posterior $q$ (Eq. 15) and the prior $p_\theta$ (Eq. 13) and parameterize $\mu_\theta(\boldsymbol{x}_t,\boldsymbol{t})$ in the prior. Since the distribution of noise is simple, we can firstly predict the added noise $\epsilon$ and use Eq. 12 and Eq. 16 to derive,

$$\mu_\theta(\boldsymbol{x}_t,\boldsymbol{t}) = \frac{1}{\sqrt{\alpha_t}}(\boldsymbol{x}_t - \frac{\beta_t}{\sqrt{1-\bar{\alpha}_t}}\boldsymbol{\epsilon}_\theta(\boldsymbol{x}_t,t))$$

(17)

And the loss function of the network can be written as,

$$L_{\text{simple}} = \mathbb{E}_{t,\boldsymbol{x}_0,\epsilon}[||\boldsymbol{\epsilon} - \boldsymbol{\epsilon}_\theta(\boldsymbol{x}_t,t)||^2]$$

(18)

At the same time, if we have trained a neural network to estimate the added noise, the recursive formulation of the reverse denoising process can be derived in closed form using Langevin dynamics Welling & Teh (2011),

$$\boldsymbol{x}_{t-1} = \frac{1}{\sqrt{\alpha_t}}(\boldsymbol{x}_t - \frac{\beta_t}{\sqrt{1-\bar{\alpha}_t}}\boldsymbol{\epsilon}_\theta(\boldsymbol{x}_t,t)) + \sigma_t\boldsymbol{z}$$

(19)

where $z \sim \mathcal{N}(0,\mathbf{I})$.

## A.2 METRICS IN THIS PAPER

We evaluated the models based on several metrics: Inception Score (IS) Salimans et al. (2016), Fréchet Inception Distance (FID) Heusel et al. (2017) and Precision-Recall(P/R) Kynkäänniemi et al. (2019). IS measures the quality and diversity of generated images, while FID measures the similarity between the generated images and real ones. To provide a thorough assessment of the generated samples' quality and mode-coverage in comparison to the training datasets, we incorporated precision-recall as an additional performance metric.

## A.3 FORMULA DERIVATION IN THE METHODS SECTION

### A.3.1 THE DERIVATION OF EQ. 4

Below is a derivation of Eq. 4, the gaussian distributions at an arbitrary timestep $t$ in the forward diffusion process. For $\boldsymbol{z}_1,\boldsymbol{z}_2 \sim \sum_{k=1}^{K}\omega_k\mathcal{N}(\boldsymbol{z};\boldsymbol{\mu}_k,\sigma_k^2\mathbf{I})$, we have

$$a\boldsymbol{z}_1 + b\boldsymbol{z}_2 \sim \sum_{i=1}^{K}\sum_{j=1}^{K}\omega_i\omega_j\mathcal{N}(\boldsymbol{z};a\boldsymbol{\mu}_i+b\boldsymbol{\mu}_j,(a^2\sigma_i^2+b^2\sigma_j^2)\mathbf{I}),$$

(20)

which indicates that the addition of two Gaussian mixture noises will produce a more complex one.

For the GMN-based diffusion process, it also admits sampling $\boldsymbol{x}_t$ at an arbitrary timestep $t$ in closed form. Let $\overline{\alpha}_t = \prod_{i=1}^{t}\alpha_i$, $\gamma_{t,i} = \beta_i\prod_{j=i+1}^{t}\alpha_j$ for $i \in [1,t-1]$, and $\gamma_{t,t} = \beta_t$, considering the

iterates of $\boldsymbol{x}_t$, we have

$$
\begin{aligned}
\boldsymbol{x}_t &= \sqrt{\alpha_t}\boldsymbol{x}_{t-1} + \sqrt{\beta_t}\boldsymbol{z}_t \\
&= \sqrt{\alpha_t\alpha_{t-1}}\boldsymbol{x}_{t-2} + \sqrt{\alpha_t\beta_{t-1}}\boldsymbol{z}_{t-1} + \sqrt{\beta_t}\boldsymbol{z}_t \\
&= ... \\
&= \sqrt{\overline{\alpha}_t}\boldsymbol{x}_0 + \sum_{i=1}^{t}\sqrt{\gamma_{t,i}}\boldsymbol{z}_i \\
&= \sqrt{\overline{\alpha}_t}\boldsymbol{x}_0 + \sqrt{\sum_{j=1}^{t}\gamma_{t,j}\sigma_{i_j}^2}\,\overline{\boldsymbol{z}}_t + \sum_{j=1}^{t}\sqrt{\gamma_{t,j}}\boldsymbol{\mu}_{i_j},
\end{aligned}
\tag{21}
$$

where $\overline{\boldsymbol{z}}_t \sim \mathcal{N}(\boldsymbol{0}, \mathbf{I})$, and $i_j \in [K]$ denotes the noise added at the $j$-the step is drawn from the $i_j$-th Gaussian distribution $\mathcal{N}(\boldsymbol{\mu}_{i_j}, \sigma_{i_j}^2\mathbf{I})$. Therefore, we have

$$
q(\boldsymbol{x}_t|\boldsymbol{x}_0) = \sum_{i_1,...,i_t\in[K]} \omega_{i_1}\ldots\omega_{i_t}\mathcal{N}\left(\boldsymbol{x}_t; \overline{\alpha}_t\boldsymbol{x}_0 + \sum_{j=1}^{t}\sqrt{\gamma_{t,j}}\boldsymbol{\mu}_{i_j}, \sum_{j=1}^{t}\gamma_{t,j}\sigma_{i_j}^2\mathbf{I}\right)
\tag{22}
$$

In order to minimize the complexity of the Gaussian mixture distribution, we set the same standard deviation for each Gaussian distribution, *i.e.*, $\sigma_1 = \cdots \sigma_K = 1$, then

$$
q(\boldsymbol{x}_t|\boldsymbol{x}_0) = \sum_{i_1,...,i_t\in[K]} \omega_{i_1}\ldots\omega_{i_t}\mathcal{N}\left(\boldsymbol{x}_t; \sqrt{\overline{\alpha}_t}\boldsymbol{x}_0 + \sum_{j=1}^{t}\sqrt{\gamma_{t,j}}\boldsymbol{\mu}_{i_j}, \sum_{j=1}^{t}\gamma_{t,j}\mathbf{I}\right)
\tag{23}
$$

This shows a same standard deviation between Gaussian distributions at an arbitrary timestep $t$.

### A.3.2 DEFINITION OF HYPERPARAMETERS IN THE FORWARD DIFFUSION PROCESS

In Eq. 4, if we define $\alpha_t = 1 - \beta_t$, then $\overline{\alpha}_t = \prod_{i=1}^{t}\alpha_i = \prod_{i=1}^{t}(1-\beta_i)$, $\gamma_{t,j} = \beta_j\prod_{k=j+1}^{t}\alpha_k = \beta_j\prod_{k=j+1}^{t}(1-\beta_k)$. Therefore, we have

$$
\gamma_t = \sum_{j=1}^{t}\gamma_{t,j} = \beta_t + \sum_{j=1}^{t-1}\beta_j\prod_{k=j+1}^{t}(1-\beta_k) = 1 - \overline{\alpha}_t = \beta_t + \alpha_t\gamma_{t-1}, \gamma_{t+1,j} = \alpha_{t+1}\gamma_{t,j}.
\tag{24}
$$

### A.3.3 THE DERIVATION OF EQ. 6

Below is a derivation of Eq. 6, the reduced variance variational bound for diffusion models.

$$
\begin{aligned}
&\mathbb{E}_{q(\boldsymbol{x}_0)}\left[-\log p_\theta(\boldsymbol{x}_0)\right] \\
=&\mathbb{E}_{q(\boldsymbol{x}_0)}\left[-\log\left(\int p_\theta(\boldsymbol{x}_{0:T})d\boldsymbol{x}_{1:T}\right)\right] \\
\leq&\mathbb{E}_{q(\boldsymbol{x}_{0:T})}\left[-\log\frac{p_\theta(\boldsymbol{x}_{0:T})}{q(\boldsymbol{x}_{1:T}|\boldsymbol{x}_0)}\right] \\
=&\mathbb{E}_{q(\boldsymbol{x}_{0:T})}\left[-\log p_\theta(\boldsymbol{x}_T) - \sum_{t\geq 1}\log p_\theta(\boldsymbol{x}_{t-1}|\boldsymbol{x}_t)\right] + C_1 \\
=&\mathbb{E}_{q(\boldsymbol{x}_{0:T})}\left[-\sum_{t>1}\mathbb{E}_{i_{1:t-1}|\boldsymbol{x}_t,\boldsymbol{x}_0}\log\frac{p_\theta(\boldsymbol{x}_{t-1}|\boldsymbol{x}_t)}{q(\boldsymbol{x}_{t-1}|\boldsymbol{x}_t,\boldsymbol{x}_0,i_{1:t-1})} - \log p_\theta(\boldsymbol{x}_T) - \log p_\theta(\boldsymbol{x}_0|\boldsymbol{x}_1)\right] + C_2
\end{aligned}
\tag{25}
$$

where

$$
\frac{p_\theta(\boldsymbol{x}_{t-1}|\boldsymbol{x}_t)}{q(\boldsymbol{x}_{t-1}|\boldsymbol{x}_t,\boldsymbol{x}_0,i_{1:t-1})} = \log\frac{\sum_{i_t\in[K]}p_\theta(i_t|\boldsymbol{x}_t)p_\theta(\boldsymbol{x}_{t-1}|\boldsymbol{x}_t,i_t)}{\sum_{i_t\in[K]}q(i_t|\boldsymbol{x}_t,\boldsymbol{x}_0,i_{1:t-1})q(\boldsymbol{x}_{t-1}|\boldsymbol{x}_t,\boldsymbol{x}_0,i_{1:t})}
\tag{26}
$$

Therefore,

$$
\mathbb{E}_{q(\boldsymbol{x}_0)}\left[-\log p_\theta(\boldsymbol{x}_0)\right]
$$

$$
\leq \mathbb{E}_{q(\boldsymbol{x}_{0:T})}\left[-\log p_\theta(\boldsymbol{x}_0|\boldsymbol{x}_1) + \sum_{t>1}\mathbb{E}_{i_{1:t-1}|\boldsymbol{x}_t,\boldsymbol{x}_0}\left[D_{\mathrm{KL}}(q(\boldsymbol{x}_{t-1}|\boldsymbol{x}_t,\boldsymbol{x}_0,i_{1:t-1})||p_\theta(\boldsymbol{x}_{t-1}|\boldsymbol{x}_t))\right]\right.
$$

$$
\left.-\log p_\theta(\boldsymbol{x}_T)\right] + C_2
$$

$$
\leq \mathbb{E}_{q(\boldsymbol{x}_{0:T})}\left[-\log p_\theta(\boldsymbol{x}_0|\boldsymbol{x}_1) + \sum_{t>1}\mathbb{E}_{i_{1:t-1}|\boldsymbol{x}_t,\boldsymbol{x}_0}\left[D_{\mathrm{KL}}(q(i_t|\boldsymbol{x}_t,\boldsymbol{x}_0,i_{1:t-1})||p_\theta(i_t|\boldsymbol{x}_t))\right]\right.
$$

$$
\left.+\sum_{t>1}\mathbb{E}_{i_{1:t}|\boldsymbol{x}_t,\boldsymbol{x}_0}\left[D_{\mathrm{KL}}(q(\boldsymbol{x}_{t-1}|\boldsymbol{x}_t,\boldsymbol{x}_0,i_{1:t})||p_\theta(\boldsymbol{x}_{t-1}|\boldsymbol{x}_t,i_t))\right] + D_{\mathrm{KL}}(q(\boldsymbol{x}_T|x_0)||p(\boldsymbol{x}_T))\right] + C_2
$$

$$
\tag{27}
$$

The last inequality is based on the fact that

$$
D_{\mathrm{KL}}(q(x)||p(x)) \leq D_{\mathrm{KL}}(q(x,y)||p(x,y)) = D_{\mathrm{KL}}(q(y)||p(y)) + D_{\mathrm{KL}}(q(x|y)||p(x|y)) \tag{28}
$$

### A.3.4 THE DERIVATION OF EQ. 9

Below is a derivation of Eq. 9, the parameterization $\boldsymbol{\mu}_\theta(\boldsymbol{x}_t,t,i_t)$. Moreover, let $\gamma_{t-1} = \sum_{j=1}^{t-1}\gamma_{t-1,j}$, we have

$$
q(\boldsymbol{x}_{t-1}|\boldsymbol{x}_t,\boldsymbol{x}_0,i_{1:t})
$$

$$
=\frac{q(\boldsymbol{x}_t|\boldsymbol{x}_{t-1},\boldsymbol{x}_0,i_{1:t})q(\boldsymbol{x}_{t-1}|\boldsymbol{x}_0,i_{1:t})}{q(\boldsymbol{x}_t|\boldsymbol{x}_0,i_{1:t})}
$$

$$
=\frac{q(\boldsymbol{x}_t|\boldsymbol{x}_{t-1},i_t)q(\boldsymbol{x}_{t-1}|\boldsymbol{x}_0,i_{1:t-1})}{q(\boldsymbol{x}_t|\boldsymbol{x}_0,i_{1:t})}
$$

$$
\propto \exp\left[-\frac{\|\boldsymbol{x}_t - \sqrt{\alpha_t}\boldsymbol{x}_{t-1} - \sqrt{\beta_t}\boldsymbol{\mu}_{i_t}\|_2^2}{2\beta_t}\right]\cdot\exp\left[-\frac{\|\boldsymbol{x}_{t-1} - \sqrt{\bar{\alpha}_{t-1}}\boldsymbol{x}_0 - \sum_{j=1}^{t-1}\sqrt{\gamma_{t-1,j}}\boldsymbol{\mu}_{i_j}\|_2^2}{2\gamma_{t-1}}\right]
$$

$$
=\exp\left[-\frac{\|\boldsymbol{x}_{t-1} - (\boldsymbol{x}_t - \sqrt{\beta_t}\boldsymbol{\mu}_{i_t})/\sqrt{\alpha_t}\|_2^2}{2\beta_t/\alpha_t}\right]\cdot\exp\left[-\frac{\|\boldsymbol{x}_{t-1} - \sqrt{\bar{\alpha}_{t-1}}\boldsymbol{x}_0 - \sum_{j=1}^{t-1}\sqrt{\gamma_{t-1,j}}\boldsymbol{\mu}_{i_j}\|_2^2}{2\gamma_{t-1}}\right]
$$

$$
\propto \exp\left[-\frac{\|\boldsymbol{x}_{t-1} - \boldsymbol{\mu}(\boldsymbol{x}_t,\boldsymbol{x}_0,i_t|i_{1:t-1})\|_2^2}{2\sigma_t^2}\right],
$$

$$
\tag{29}
$$

where

$$
\boldsymbol{\mu}(\boldsymbol{x}_t,\boldsymbol{x}_0,i_t|i_{1:t-1})
$$

$$
=\frac{1}{\beta_t/\alpha_t + \gamma_{t-1}}\left[\gamma_{t-1}(\boldsymbol{x}_t - \sqrt{\beta_t}\boldsymbol{\mu}_{i_t})/\sqrt{\alpha_t} + \beta_t/\alpha_t\left(\sqrt{\bar{\alpha}_{t-1}}\boldsymbol{x}_0 + \sum_{j=1}^{t-1}\sqrt{\gamma_{t-1,j}}\boldsymbol{\mu}_{i_j}\right)\right]
$$

$$
=\frac{1}{\beta_t + \alpha_t\gamma_{t-1}}\left[\gamma_{t-1}\sqrt{\alpha_t}(\boldsymbol{x}_t - \sqrt{\beta_t}\boldsymbol{\mu}_{i_t}) + \beta_t\left(\sqrt{\bar{\alpha}_{t-1}}\boldsymbol{x}_0 + \sum_{j=1}^{t-1}\sqrt{\gamma_{t-1,j}}\boldsymbol{\mu}_{i_j}\right)\right]
$$

$$
=\frac{1}{\beta_t + \alpha_t\gamma_{t-1}}\left[\gamma_{t-1}\sqrt{\alpha_t}\boldsymbol{x}_t + \beta_t\sqrt{\bar{\alpha}_{t-1}}\boldsymbol{x}_0 - \gamma_{t-1}\sqrt{\alpha_t\beta_t}\boldsymbol{\mu}_{i_t} + \beta_t\sum_{j=1}^{t-1}\sqrt{\gamma_{t-1,j}}\boldsymbol{\mu}_{i_j}\right]
$$

$$
\tag{30}
$$

$$
\sigma_t^2 = \frac{\beta_t/\alpha_t\gamma_{t-1}}{\beta_t/\alpha_t + \gamma_{t-1}} = \frac{\beta_t\gamma_{t-1}}{\beta_t + \alpha_t\gamma_{t-1}} = \frac{\beta_t\gamma_{t-1}}{\gamma_t} \quad (\text{if } \alpha_t = 1 - \beta_t)
$$

Let $\overline{\boldsymbol{\mu}} = \mathbb{E}_{i_j} \boldsymbol{\mu}_{i_j} = \sum_{i=1}^{K} \omega_k \boldsymbol{\mu}_k$, and $\Sigma_\theta(\boldsymbol{x}_t, t, k) = \sigma_t^2(k)\mathbf{I}$, thus we have

$$\mathbb{E}_{i_{1:t}|\boldsymbol{x}_t,\boldsymbol{x}_0} D_{\mathrm{KL}}(q(\boldsymbol{x}_{t-1}|\boldsymbol{x}_t,\boldsymbol{x}_0,i_{1:t})||p_\theta(\boldsymbol{x}_{t-1}|\boldsymbol{x}_t,i_t))$$

$$=\mathbb{E}_{i_{1:t}|\boldsymbol{x}_t,\boldsymbol{x}_0} \left[\log\frac{\sigma_t(i_t)}{\sigma_t} + \frac{\sigma_t^2 + \|\boldsymbol{\mu}(\boldsymbol{x}_t,\boldsymbol{x}_0,i_t|i_{1:t-1}) - \boldsymbol{\mu}_\theta(\boldsymbol{x}_t,t,i_t)\|_2^2}{2\sigma_t^2(i_t)}\right]$$

$$=\mathbb{E}_{i_t|\boldsymbol{x}_t,\boldsymbol{x}_0} \left[\log\frac{\sigma_t(i_t)}{\sigma_t} + \frac{\sigma_t^2 + \mathbb{E}_{i_{1:t-1}|\boldsymbol{x}_t,\boldsymbol{x}_0,i_t}\|\boldsymbol{\mu}(\boldsymbol{x}_t,\boldsymbol{x}_0,i_t|i_{1:t-1}) - \boldsymbol{\mu}_\theta(\boldsymbol{x}_t,t,i_t)\|_2^2}{2\sigma_t^2(i_t)}\right] \quad (31)$$

$$=\mathbb{E}_{i_t|\boldsymbol{x}_t,\boldsymbol{x}_0} \left[\log\frac{\sigma_t(i_t)}{\sigma_t} + \frac{\sigma_t^2 + \|\mathbb{E}_{i_{1:t-1}|\boldsymbol{x}_t,\boldsymbol{x}_0,i_t}\boldsymbol{\mu}(\boldsymbol{x}_t,\boldsymbol{x}_0,i_t|i_{1:t-1}) - \boldsymbol{\mu}_\theta(\boldsymbol{x}_t,t,i_t)\|_2^2}{2\sigma_t^2(i_t)}\right] + C$$

For brevity, let

$$\tilde{\boldsymbol{\mu}}(\boldsymbol{x}_t,\boldsymbol{x}_0,i_t) = \mathbb{E}_{i_{1:t-1}|\boldsymbol{x}_t,\boldsymbol{x}_0,i_t}\boldsymbol{\mu}(\boldsymbol{x}_t,\boldsymbol{x}_0,i_{1:t-1}), \quad (32)$$

then we can derive that

$$\tilde{\boldsymbol{\mu}}(\boldsymbol{x}_t,\boldsymbol{x}_0,i_t)$$

$$=\mathbb{E}_{i_{1:t-1}|\boldsymbol{x}_t,\boldsymbol{x}_0,i_t}\boldsymbol{\mu}(\boldsymbol{x}_t,\boldsymbol{x}_0,i_t|i_{1:t-1})$$

$$= \int q(i_{1:t-1}|\boldsymbol{x}_t,\boldsymbol{x}_0,i_t)\boldsymbol{\mu}(\boldsymbol{x}_t,\boldsymbol{x}_0,i_t|i_{1:t-1})\mathrm{d}i_{1:t-1}$$

$$= \int \frac{q(\boldsymbol{x}_t|\boldsymbol{x}_0,i_{1:t})q(i_{1:t-1})}{q(\boldsymbol{x}_t|\boldsymbol{x}_0,i_t)}\boldsymbol{\mu}(\boldsymbol{x}_t,\boldsymbol{x}_0,i_t|i_{1:t-1})\mathrm{d}i_{1:t-1}$$

$$=\frac{\gamma_{t-1}\sqrt{\alpha_t}\boldsymbol{x}_t + \beta_t\sqrt{\overline{\alpha}_{t-1}}\boldsymbol{x}_0 - \gamma_{t-1}\sqrt{\alpha_t\beta_t}\boldsymbol{\mu}_{i_t}}{\beta_t + \alpha_t\gamma_{t-1}} + \frac{\beta_t\sum_{j=1}^{t-1}\sqrt{\gamma_{t-1,j}}\int q(\boldsymbol{x}_t|\boldsymbol{x}_0,i_t,i_j)q(i_j)\boldsymbol{\mu}_{i_j}\mathrm{d}i_j}{q(\boldsymbol{x}_t|\boldsymbol{x}_0,i_t)(\beta_t + \alpha_t\gamma_{t-1})}$$

$$=\frac{\gamma_{t-1}\sqrt{\alpha_t}\boldsymbol{x}_t + \beta_t\sqrt{\overline{\alpha}_{t-1}}\boldsymbol{x}_0 - \gamma_{t-1}\sqrt{\alpha_t\beta_t}\boldsymbol{\mu}_{i_t}}{\beta_t + \alpha_t\gamma_{t-1}} +$$

$$\frac{\beta_t}{\beta_t + \alpha_t\gamma_{t-1}}\sum_{j=1}^{t-1}\sqrt{\gamma_{t-1,j}}\int \frac{q(i_j)q(\boldsymbol{x}_t|\boldsymbol{x}_0,i_t,i_j)}{q(\boldsymbol{x}_t|\boldsymbol{x}_0,i_t)}\boldsymbol{\mu}_{i_j}\mathrm{d}i_j,$$

$$(33)$$

and

$$\tilde{\sigma}_t^2(i_t) = \sigma_t^2 + \mathbb{E}_{i_{1:t-1}|\boldsymbol{x}_t,\boldsymbol{x}_0,i_t}\|\boldsymbol{\mu}(\boldsymbol{x}_t,\boldsymbol{x}_0,i_t|i_{1:t-1}) - \tilde{\boldsymbol{\mu}}(\boldsymbol{x}_t,\boldsymbol{x}_0,i_t)\|_2^2$$

$$= \sigma_t^2 + \frac{\beta_t^2}{(\beta_t + \alpha_t\gamma_{t-1})^2}\left[\mathbb{E}_{i_{1:t-1}|\boldsymbol{x}_t,\boldsymbol{x}_0,i_t}\left\|\sum_{j=1}^{t-1}\sqrt{\gamma_{t-1,j}}\boldsymbol{\mu}_{i_j}\right\|_2^2 - \right.$$

$$\left.\left\|\mathbb{E}_{i_{1:t-1}|\boldsymbol{x}_t,\boldsymbol{x}_0,i_t}\sum_{j=1}^{t-1}\sqrt{\gamma_{t-1,j}}\boldsymbol{\mu}_{i_j}\right\|_2^2\right] \quad (34)$$

$$\approx \sigma_t^2 + \frac{\beta_t^2}{(\beta_t + \alpha_t\gamma_{t-1})^2}\left[\gamma_{t-1}\mathbb{E}_k\|\boldsymbol{\mu}_k\|_2^2 + [(\sum_{j=1}^{t-1}\sqrt{\gamma_{t-1,j}})^2 - \gamma_{t-1}]\|\mathbb{E}_k\boldsymbol{\mu}_k\|_2^2\right]$$

To represent the mean $\boldsymbol{\mu}_\theta(\boldsymbol{x}_t,t)$, we propose a specific parameterization motivated by the following analysis. We can expand the above equation further by reparameterizing $\boldsymbol{x}_t(\boldsymbol{x}_0,\boldsymbol{\epsilon},i_1,...,i_t) = \sqrt{\overline{\alpha}_t}\boldsymbol{x}_0 + \sqrt{\gamma_t}\boldsymbol{\epsilon} + \sum_{j=1}^{t}\sqrt{\gamma_{t,j}}\boldsymbol{\mu}_{i_j}$, i.e., $\boldsymbol{x}_0 = \frac{1}{\sqrt{\overline{\alpha}_t}}\left(\boldsymbol{x}_t - \sqrt{\gamma_t}\boldsymbol{\epsilon} - \sum_{j=1}^{t}\sqrt{\gamma_{t,j}}\boldsymbol{\mu}_{i_j}\right)$, then we have

$$\boldsymbol{\mu}(\boldsymbol{x}_t,\boldsymbol{x}_0,i_t|i_{1:t-1})$$

$$=\frac{\gamma_{t-1}\sqrt{\alpha_t}\boldsymbol{x}_t + \beta_t\sqrt{\overline{\alpha}_{t-1}}\boldsymbol{x}_0 - \gamma_{t-1}\sqrt{\alpha_t\beta_t}\boldsymbol{\mu}_{i_t} + \beta_t\sum_{j=1}^{t-1}\sqrt{\gamma_{t-1,j}}\boldsymbol{\mu}_{i_j}}{\beta_t + \alpha_t\gamma_{t-1}}$$

$$=\frac{\gamma_{t-1}\sqrt{\alpha_t}\boldsymbol{x}_t + \frac{\beta_t}{\sqrt{\alpha_t}}\left(\boldsymbol{x}_t - \sqrt{\gamma_t}\boldsymbol{\epsilon} - \sum_{j=1}^{t}\sqrt{\gamma_{t,j}}\boldsymbol{\mu}_{i_j}\right) - \gamma_{t-1}\sqrt{\alpha_t\beta_t}\boldsymbol{\mu}_{i_t} + \beta_t\sum_{j=1}^{t-1}\sqrt{\gamma_{t-1,j}}\boldsymbol{\mu}_{i_j}}{\beta_t + \alpha_t\gamma_{t-1}}$$

$$=\frac{1}{\sqrt{\alpha_t}}\frac{(\beta_t + \alpha_t\gamma_{t-1})\boldsymbol{x}_t - \beta_t\sqrt{\gamma_t}\boldsymbol{\epsilon} - \sqrt{\beta_t}(\beta_t + \alpha_t\gamma_{t-1})\boldsymbol{\mu}_{i_t} + \beta_t\sum_{j=1}^{t-1}(\sqrt{\alpha_t\gamma_{t-1,j}} - \sqrt{\gamma_{t,j}})\boldsymbol{\mu}_{i_j}}{\beta_t + \alpha_t\gamma_{t-1}}$$

$$=\frac{1}{\sqrt{\alpha_t}}\left(\boldsymbol{x}_t - \sqrt{\beta_t}\boldsymbol{\mu}_{i_t} - \frac{\beta_t\sqrt{\gamma_t}}{\beta_t + \alpha_t\gamma_{t-1}}\boldsymbol{\epsilon}\right)$$

$$(35)$$

---

**Algorithm 1** Training

---
1: **repeat**
2:     $x_0 \sim q(x_0)$, $t \sim \text{Uniform}(\{1, ..., T\})$
3:     $\epsilon \sim \mathcal{N}(\mathbf{0}, \mathbf{I})$, $i_1, ..., i_t \sim \text{Uniform}(\{1, ..., K\})$
4:     $x_t = \sqrt{\bar{\alpha}_t} x_0 + \sqrt{\gamma_t} \epsilon + \sum_{j=1}^{t} \sqrt{\gamma_{t,j}} \boldsymbol{\mu}_{i_j}$
5:     gradient descent step on

$$\nabla_\theta \left( \frac{\gamma_t \beta_t^2}{2\tilde{\sigma}_t^2 \alpha_t (\beta_t + \alpha_t \gamma_{t-1})^2} \|\epsilon - \epsilon_\theta(x_t)\|^2 - \log p_\theta(i_t | x_t) \right)$$

6: **until** converged

---

**Algorithm 2** Standard Sampling

---
1: $x_T \sim \mathcal{N}\left(0, \mathbf{I} + \gamma_T \mathbb{E}[\boldsymbol{\mu}_k \boldsymbol{\mu}_k^\top]\right)$, $T$
2: **for** $t = T, ..., 1$ **do**
3:     $z \sim \mathcal{N}(0, \mathbf{I})$ if $t > 1$, else $x = \mathbf{0}$
4:     $i_t = \arg\max_k p_\theta(k | x_t)$
5:     $x_{t-1} = \frac{1}{\sqrt{\alpha_t}} \left( x_t - \sqrt{\beta_t} \boldsymbol{\mu}_{i_t} - \frac{\beta_t \sqrt{\gamma_t}}{\beta_t + \alpha_t \gamma_{t-1}} \epsilon_\theta(x_t, t) \right)$
     $+ \tilde{\sigma}_t(i_t) z$
6: **end for**
7: **Return** $x_0$

---

The above equation reveals that $\boldsymbol{\mu}_\theta$ must predict $\frac{1}{\sqrt{\alpha_t}} \left( x_t - \sqrt{\beta_t} \boldsymbol{\mu}_{i_t} - \frac{\beta_t \sqrt{\gamma_t}}{\beta_t + \alpha_t \gamma_{t-1}} \epsilon \right)$ given $x_t$. Since $x_t$ is available as input to the model, we may choose the parameterization $\boldsymbol{\mu}_\theta(x_t, t, i_t)$

$$\frac{1}{\sqrt{\alpha_t}} \left( x_t - \sqrt{\beta_t} \boldsymbol{\mu}_{i_t} - \frac{\beta_t \sqrt{\gamma_t}}{\beta_t + \alpha_t \gamma_{t-1}} \epsilon_\theta(x_t, t) \right), \tag{36}$$

where $\epsilon_\theta$ is a function approximation intended to predict $\epsilon$ from $x_t$.

### A.4 DETAILED ALGORITHM

We show the details of the algorithm in the paper. Algorithm 1 shows the complete training procedure. At the same time, by using Langevin dynamics Welling & Teh (2011), we can gradually sample the image from Gaussian mixture noise in Algorithm 2. We also propose three different sampling strategies to simplify the sampling process. Among them, the first strategy, directly utilizes the original DDPM sampler Ho et al. (2020) as a foundation (shown in Algorithm 3). In Eq. 3, in addition to the terms involving $x_0$ and the noise $\bar{z}t$, which are similar to the original DDPM, we introduce an additional term representing the mean value $\boldsymbol{\mu} i_j$. From a global perspective, we can know from the central limit theorem that it is equivalent to introducing a Gaussian noise globally, augmenting the original DDPM in the training phase. This augmentation enhances the model's expressiveness by incorporating additional noise sources, and theoretically, it can achieve better results by using the original DDPM sampler directly in the sampling phase. The second strategy involves directly sampling the image removing the mean value $\boldsymbol{\mu}$, which is shown in Algorithm 4. As we can see, we have made adjustments to the initialization sampling step based on the first strategy, enhancing its effectiveness and adaptability. And the third strategy involves randomly sampling the mean value $\boldsymbol{\mu}$ according to its probability distribution and then using standard sampling to generate the image. The algorithm of the third strategy is shown in Algorithm 5.

### A.5 MORE RESULTS

In this section, we show more subjective results generated by our GMN-DDPM.

---

**Algorithm 3** Sampling using original DDPM sampling method

---

1: $\boldsymbol{x}_T \sim \mathcal{N}\left(0, \mathbf{I}\right), T$
2: **for** t=T,...,1 **do**
3:      $\boldsymbol{z} \sim \mathcal{N}(0, \mathbf{I})$ if $t > 1$, else $\boldsymbol{x} = \mathbf{0}$
4:      $\boldsymbol{x}_{t-1} = \frac{1}{\sqrt{\alpha_t}}\left(\boldsymbol{x}_t - \frac{\beta_t\sqrt{\gamma_t}}{\beta_t + \alpha_t\gamma_{t-1}}\boldsymbol{\epsilon}_\theta(\boldsymbol{x}_t, t)\right) + \tilde{\sigma}_t(i_t)\boldsymbol{z}$
5: **end for**
6: **Return** $\boldsymbol{x}_0$

---

**Algorithm 4** Sampling without the removal of $\boldsymbol{\mu}$

---

1: $\boldsymbol{x}_T \sim \mathcal{N}\left(0, \mathbf{I} + \gamma_T\mathbb{E}[\boldsymbol{\mu}_k\boldsymbol{\mu}_k^\top]\right), T$
2: **for** t=T,...,1 **do**
3:      $\boldsymbol{z} \sim \mathcal{N}(0, \mathbf{I})$ if $t > 1$, else $\boldsymbol{x} = \mathbf{0}$
4:      $\boldsymbol{x}_{t-1} = \frac{1}{\sqrt{\alpha_t}}\left(\boldsymbol{x}_t - \frac{\beta_t\sqrt{\gamma_t}}{\beta_t + \alpha_t\gamma_{t-1}}\boldsymbol{\epsilon}_\theta(\boldsymbol{x}_t, t)\right) + \tilde{\sigma}_t(i_t)\boldsymbol{z}$
5: **end for**
6: **Return** $\boldsymbol{x}_0$

---

**Algorithm 5** Sampling using random $\boldsymbol{\mu}$

---

1: $\boldsymbol{x}_T \sim \mathcal{N}\left(0, \mathbf{I} + \gamma_T\mathbb{E}[\boldsymbol{\mu}_k\boldsymbol{\mu}_k^\top]\right), T$
2: **for** t=T,...,1 **do**
3:      $\boldsymbol{z} \sim \mathcal{N}(0, \mathbf{I})$ if $t > 1$, else $\boldsymbol{x} = \mathbf{0}$
4:      $i_t$ is randomly sampled by probability
5:      $\boldsymbol{x}_{t-1} = \frac{1}{\sqrt{\alpha_t}}\left(\boldsymbol{x}_t - \sqrt{\beta_t}\boldsymbol{\mu}_{i_t} - \frac{\beta_t\sqrt{\gamma_t}}{\beta_t + \alpha_t\gamma_{t-1}}\boldsymbol{\epsilon}_\theta(\boldsymbol{x}_t, t)\right)$
      $+ \tilde{\sigma}_t(i_t)\boldsymbol{z}$
6: **end for**
7: **Return** $\boldsymbol{x}_0$

---

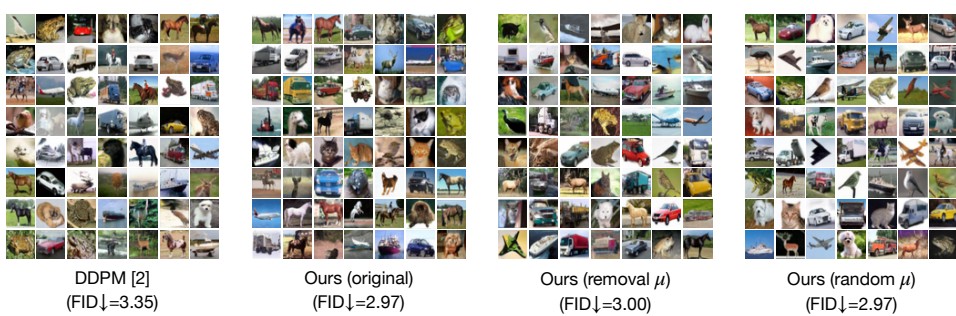

DDPM [2]
(FID↓=3.35)

Ours (original)
(FID↓=2.97)

Ours (removal $\mu$)
(FID↓=3.00)

Ours (random $\mu$)
(FID↓=2.97)

Figure 6: Several images generated on CIFAR-10 Krizhevsky et al. (2009) dataset ($32\times32$) by using the original DDPM and our GMN-DDPM.

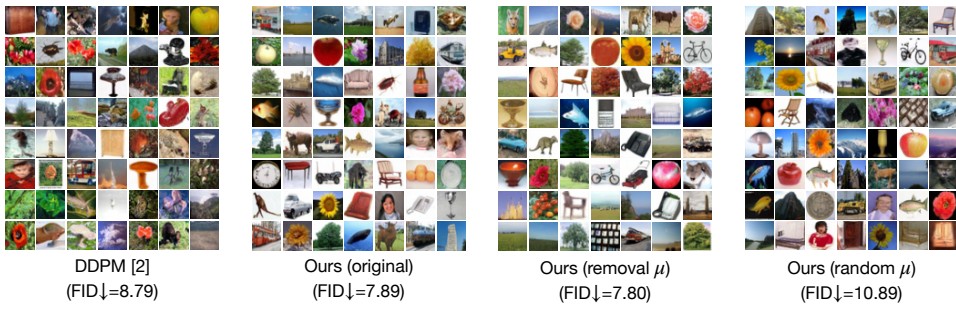

| DDPM [2] | Ours (original) | Ours (removal $\mu$) | Ours (random $\mu$) |
|---|---|---|---|
| (FID↓=8.79) | (FID↓=7.89) | (FID↓=7.80) | (FID↓=10.89) |

Figure 7: Several images generated on CIFAR-100 Krizhevsky et al. (2009) dataset ($32 \times 32$) by using the original DDPM and our GMN-DDPM.

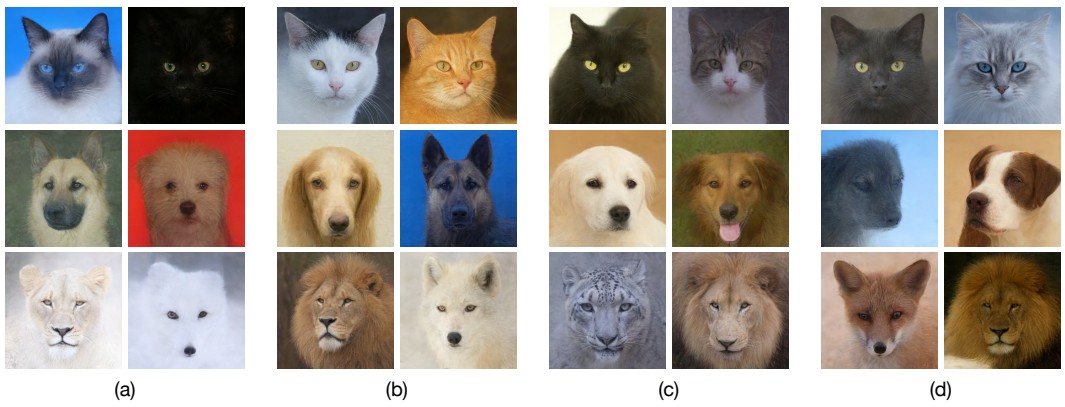

(a)      (b)      (c)      (d)

Figure 8: Several images generated on AFHQ-v2 Choi et al. (2020) dataset ($256 \times 256$) by using the original DDPM and our GMN-DDPM.

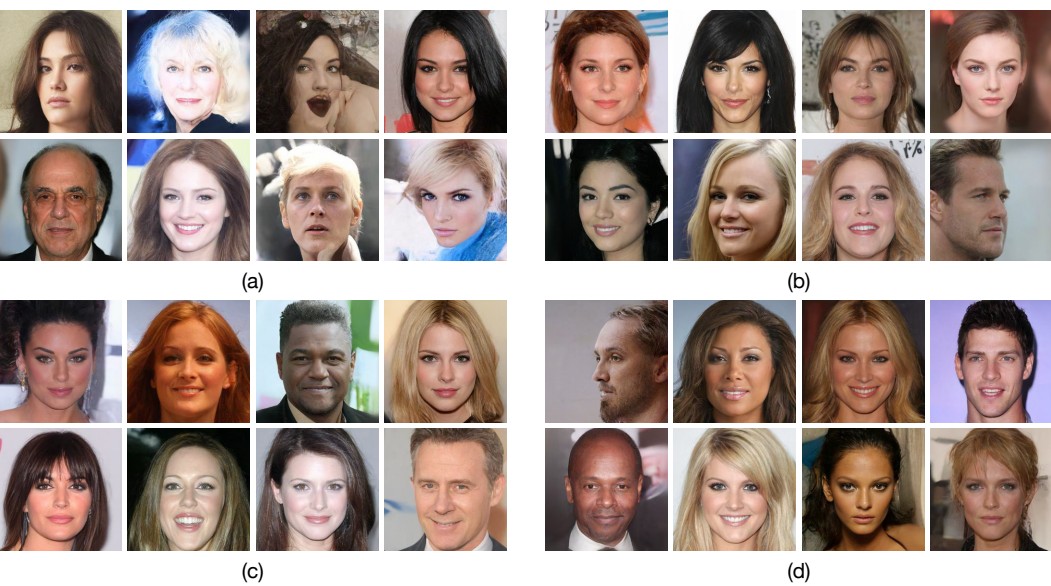

(a)      (b)

(c)      (d)

Figure 9: Several images generated on CelebA Liu et al. (2015) dataset ($256 \times 256$) by using the original DDPM and our GMN-DDPM.

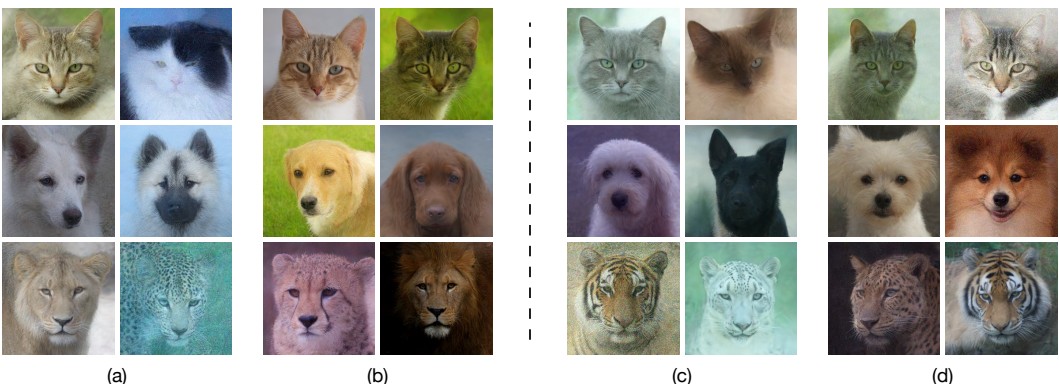

(a)        (b)        (c)        (d)

Figure 10: Several images generated on AFHQ-v2 Choi et al. (2020) dataset ($256 \times 256$) by using the fast dedicated solvers for diffusion ODEs including DDIM Song et al. (2020) and dpm-solver Lu et al. (2022).

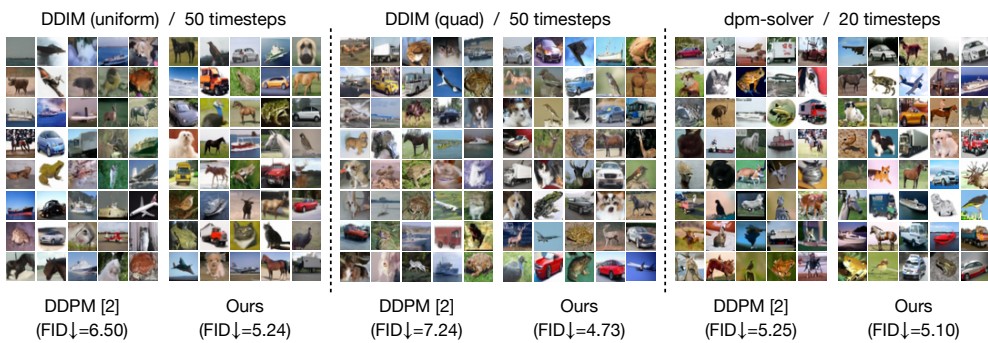

Figure 11: Several images generated on CIFAR-10 Krizhevsky et al. (2009) dataset ($32 \times 32$) by using the fast dedicated solvers for diffusion ODEs including DDIM Song et al. (2020) and dpm-solver Lu et al. (2022).

