# OpenReview forum: "GM-DDPM: Denoising diffusion probabilistic models with Gaussian Mixture Noise"
_ICLR.cc/2024/Conference — ICLR 2024 Conference Withdrawn Submission_

### Official Review · Reviewer_JGQA · 2023-10-26

**Soundness:** 2 fair
**Presentation:** 2 fair
**Contribution:** 2 fair
**Rating:** 5
**Confidence:** 4

**Summary:**

This paper tried to improve the denoising diffusion probabilistic models (DDPMs) by parameterizing the noise in the forward diffusion process with a more flexible form, i.e., the Gaussian mixture. As claimed by the authors, such a Gaussian mixture noise can make the image more quickly degrades to pure noise in the forward diffusion process, and thus facilitates the reverse sampling process for higher-quality image generation. Experiments have been conducted to demonstrate the effectiveness of the proposed method.

**Strengths:**

1. The authors have deeply investigated the mechanism of DDPM, and provide new thoughts for DDPMs

2. Experimental results reveal some advantages of the proposed method.

**Weaknesses:**

First, the reasonability of the proposed method is not convincing enough. The authors claimed that the fast degradation in the forward diffusion process can facilitate a better generation. However, the mechanism underlying this claim is unclear, and the empirical differences between the new method and the original DDPM are not significant, as I can see from the experimental results.

Second, the experiments are not sufficient. Specifically, the authors have included many methods not within the DDPM class, while only compared with the original DDPM. To my knowledge, there are many variants of DDPM currently, and several of them considered to better parameterize the distribution in the reverse denoising process, which should be compared.

**Questions:**

In addition to the weaknesses mentioned above, I doubt that the better generation results might be due to the parameterization of the distribution in the reverse process, which is also a Gaussian mixture according to Eq. (5). Therefore, it is better to consider to variants of the proposed method, i.e., Gaussian noise in forward process together with Gaussian mixture in reverse process, and Gaussian mixture noise in forward process together with Gaussian in reverse process. With such ablations, we may observe whether the forward distribution is as important as the authors claimed.

---

### Official Review · Reviewer_NA66 · 2023-10-30

**Soundness:** 2 fair
**Presentation:** 2 fair
**Contribution:** 2 fair
**Rating:** 5
**Confidence:** 3

**Summary:**

This paper proposed an extended version of DDPM that add noises with Gaussian mixture noises. The paper presented the reformulated forward and reverse formulation of the proposed model. It also presents the derived training loss that is based on the variational inference, as an extension to mixture of Gaussian noises. The experimental results compared with the baseline diffusion models, and shows marginal improvements in the generated image quality.

**Strengths:**

The idea to generalize the Gaussian noise in diffusion model to the Gaussian mixture model is the key idea, which seems to be interesting. The paper derived the forward and reverse process of the revised diffusion model, and the derived sampling model relies on a classification on the Gaussian component to be chosen.  The experimental results, though I have concerns as follows, show improvements in quality of generated image on benchmark datasets.

**Weaknesses:**

1. The motivation on the extension of noises from Gaussian to Mixture of Gaussian for diffusion model is not very clear. The paper utilized Gaussian components in different means and fixed variance. It may introduce global bias(means) in different steps, but why this can introduce improvement for generating images?

2. The details on the classification (for the chosen of Gaussian component) are not clear. How to design and train this classification network, and it outputs probabilities to different Gaussian components? In the reverse process, in each step, whether a single Gaussian component is chosen, and why?

3. In experiments, it maybe better to show the chosen Gaussian components for better understanding the reverse sampling process. Are the forward and backward process using the same Gaussian component in a same corresponding forward and reverse step, e.g., t?

4. The compared diffusion models are published before 2020 (see Table 1), the paper should compare with more recent Sota diffusion models to see the position of this proposed method.

5. In the visual comparison results, it is hard to see the difference to the compared methods. What is the advantage of the proposed approach in visual comparisons? In Figs. 8-10, it is not clear on the corresponding methods for the presented generated images.

**Questions:**

Please see my above concerns on motivations, implementation/training details, and comparisons.

---

### Official Review · Reviewer_mGnb · 2023-10-31

**Soundness:** 2 fair
**Presentation:** 2 fair
**Contribution:** 3 good
**Rating:** 5
**Confidence:** 2

**Summary:**

This paper improves the denoising diffusion probabilistic model by introducing Gaussian mixture noise instead of single Gaussian noise, and presents three sampling strategies to boost the efficacy of image representation.

**Strengths:**

The ideal of using mixture of Gaussian instead of a single Gaussian in diffusion model is interesting.

**Weaknesses:**

1.	There are some related works which also use Gaussian mixture noise in DDPM such as [A], the authors should discuss and compare the current work with the existing work.
[A] Nachmani, Eliya, Robin San Roman, and Lior Wolf. "Non gaussian denoising diffusion models." arXiv preprint arXiv:2106.07582 (2021).
2.	The authors introduce the GM-DDPM model, specifically the forward diffusion process and the reverse denoising process, while there is no theoretical guarantee to indicate that the proposed GM-DDPM is superior to the original DDPM. The authors only show the efficiency of GM-DDPM in section 3.3 with some limited empirical evidence.
3.	The experimental results do not show a significant improvement of the GM-DDPM over the original DDPM. Specifically, the advantages of using Gaussian mixture noise over single Gaussian noise are not fully exhibited both theory and experiments.
4.	The authors argue that “the limitation of original diffusion models lie in their use of a single Gaussian distribution, which restricts their expressive ability…”. Can you explain this in more details?
5.	In Figure 2, it is not clear which results are from original DDPM and which are results from the proposed method.
6.	In section 3.3, the authors aim to show that GM-DDPM exhibits a faster degradation of the images. While I cannot fully understand why does faster image degradation is good for the training of the diffusion model.

**Questions:**

Please refer to the Weakness part.

---

### Official Review · Reviewer_uDWr · 2023-10-31

**Soundness:** 2 fair
**Presentation:** 1 poor
**Contribution:** 1 poor
**Rating:** 3
**Confidence:** 4

**Summary:**

This paper presented GM-DDPM, a novel paradigm that extends original diffusion models by incorporating Gaussian mixture noise. By introducing multiple Gaussian components, GM-DDPM facilitates faster and more effective degradation of image structure during the diffusion process. This augmentation, combined with the proposed sampling strategies and the utilization of fast dedicated solvers, improves the efficiency and accuracy of distribution estimation.

**Strengths:**

This paper proposed a new DDPM mode by replacing the single Gaussian noise with the GMM noise.

**Weaknesses:**

1. The motivation for introducing the GMM into DDPM is not convincing.

2. The novelty of this work is limited since previous work has used GMM in DDPM.

**Questions:**

1. The proposed GM-DDPM is actually a diffusion model (DM) that adds Gaussian mixture noise (GMN) instead of single Gaussian noise to destroy images in each diffusion step. The motivation for introducing GMN to destroy data samples is not convincing. In my opinion, the forward process of DMs only needs to map the unknown data distribution to a known distribution (e.g. normal distribution), and the reverse process learns the reversed map. The single Gaussian noise-based DMs can achieve this goal well. Besides, incorporating  GMN may increase the training difficulty. Therefore, it is not necessary to use the GMN to modify the original DDPM.

2. In sec. 3.3, the paper claims that GM-DDPM has superior efficiency of noise diffusion and denoising processes compared to the original DDPM with respect to the PSNR and classification of images in the diffusion process. The concern is whether DDPM yields similar results when the mean of added Gaussian distribution is non-zero. And whether DDPM-based single Gaussian noise with non-zero mean can enhance the expressive ability for complex image distributions.

3. The novelty of this paper is limited. As far as I know, there are two related works that introduce GMN to DDPM [1][2]. The authors should explain the differences between the proposed method and these two methods,  and I would like to see the performance comparison with them.

[1] Nachmani et al. Non Gaussian Denoising Diffusion Models, 2021
[2] Ma et al. Approximated Anomalous Diffusion: Gaussian Mixture Score-based Generative Models, 2022